# The Chd1 chromatin remodeler shifts hexasomes unidirectionally

Robert F Levendosky[1], Anton Sabantsev[2], Sebastian Deindl[2], Gregory D Bowman[1]*

[1]T.C. Jenkins Department of Biophysics, Johns Hopkins University, Baltimore, United States; [2]Department of Cell and Molecular Biology, Science for Life Laboratory, Uppsala University, Uppsala, Sweden

**Abstract** Despite their canonical two-fold symmetry, nucleosomes in biological contexts are often asymmetric: functionalized with post-translational modifications (PTMs), substituted with histone variants, and even lacking H2A/H2B dimers. Here we show that the Widom 601 nucleosome positioning sequence can produce hexasomes in a specific orientation on DNA, providing a useful tool for interrogating chromatin enzymes and allowing for the generation of nucleosomes with precisely defined asymmetry. Using this methodology, we demonstrate that the Chd1 chromatin remodeler from *Saccharomyces cerevisiae* requires H2A/H2B on the entry side for sliding, and thus, unlike the back-and-forth sliding observed for nucleosomes, Chd1 shifts hexasomes unidirectionally. Chd1 takes part in chromatin reorganization surrounding transcribing RNA polymerase II (Pol II), and using asymmetric nucleosomes we show that ubiquitin-conjugated H2B on the entry side stimulates nucleosome sliding by Chd1. We speculate that biased nucleosome and hexasome sliding due to asymmetry contributes to the packing of arrays observed in vivo.

*For correspondence:
gdbowman@jhu.edu

Competing interests: The authors declare that no competing interests exist.

## Introduction

As the repeating unit of chromatin, the nucleosome is the canvas upon which the epigenetic histone code is written. A fundamental characteristic of the histone code is the combinatorial diversity achieved from multiple marks, which may or may not reside on the same histone tail (*Ruthenburg et al., 2007*; *Tee and Reinberg, 2014*). Both through post-translational modifications (PTMs) and substitution of histone variants, additional chemical diversity arises from asymmetric modifications of nucleosomes. Since the nucleosome is pseudo-symmetric with two copies of each core histone (H2A, H2B, H3 and H4), asymmetry occurs when each copy possesses distinct epigenetic modifications. Recent advances have revealed asymmetry at the single nucleosome level (*Rhee et al., 2014*; *Voigt et al., 2012*), yet with challenges in synthesizing uniform populations of asymmetrically modified nucleosomes (*Lechner et al., 2016*; *Liokatis et al., 2016*), the biological significance of the vast majority of asymmetric marks remains unclear.

A dramatic example of asymmetry is the pairing of activating H3K4me3 and repressive H3K27me3 marks, known as bivalency (*Voigt et al., 2013*). Trimethylation of H3K27 is carried out by PRC2, and while H3K4me3 blocks modification of K27 on the same H3 tail, PRC2 can deposit a H3K27me3 mark on the opposing H3 tail of the same nucleosome (*Lechner et al., 2016*; *Voigt et al., 2012*). In addition to generating nucleosomes with asymmetric H3K4me3/H3K27me3, PRC2 is also activated by the mark it deposits, with substrate preference for asymmetric nucleosomes containing one H3K27me3 (*Lechner et al., 2016*; *Margueron et al., 2009*). While recognition of asymmetric H3K27me3 is believed to be important for maintenance and spreading of heterochromatin, and the bivalent H3K4me3/H3K27me3 signature has been well established for stem cell

identity, there is relatively little biological understanding for most other epigenetic marks that are prominently asymmetric. Genome-wide studies have revealed that the +1 nucleosome is strikingly asymmetric with regards to H3K9 acetylation, H2B ubiquitination, and residency of H2A.Z (*Rhee et al., 2014*). Asymmetric marks of the +1 nucleosome correlate with asymmetric localization of the RSC, INO80, and SWR1 chromatin remodelers (*Ramachandran et al., 2015*; *Yen et al., 2012*), and a major question is how these and other enzymes generate and read-out the asymmetric distribution of these marks.

Nucleosomes can also exhibit asymmetry with respect to histone content, with the lack of one H2A/H2B dimer defining the hexasome. The existence of hexasomes in vivo has been supported by ChIP-exo and MNase-seq experiments (*Rhee et al., 2014*), and in vitro, hexasomes have been shown to be generated by the RSC remodeler with the NAP1 histone chaperone (*Kuryan et al., 2012*) and also by RNA polymerase II (Pol II) transcribing through nucleosomes (*Kireeva et al., 2002*, *2005*). Intriguingly, Pol II successfully transcribes through hexasomes oriented with the promoter-distal H2A/H2B dimer missing, but stalls in the absence of the promoter-proximal dimer (*Kulaeva et al., 2009*). Whether the orientation of hexasomes may affect other enzymes that act on chromatin has not previously been addressed.

Transcription requires local disruption and reassembly of nucleosomes, which is achieved by elongation factors, histone chaperones, and chromatin remodelers such as Chd1 and ISWI (*Venkatesh and Workman, 2015*). Chd1 and ISWI reposition nucleosomes into evenly spaced arrays, and are required for packing arrays of nucleosomes against the +1 nucleosome (*Gkikopoulos et al., 2011*; *Lusser et al., 2005*; *Pointner et al., 2012*; *Tsukiyama et al., 1999*). Although specific binding of H3K4me3 by the chromodomains of mouse Chd1 has been correlated with its localization to the promoter (*Lin et al., 2011*), Chd1 and ISWI remodelers have been shown to participate in resetting the chromatin barrier in coding regions after passage of Pol II, required for preventing cryptic transcription (*Cheung et al., 2008*; *Pointner et al., 2012*; *Radman-Livaja et al., 2012*; *Smolle et al., 2012*). Chd1 has been linked to elongating Pol II through interactions with the transcriptional elongation factors FACT and Spt4-Spt5, and with the Rtf1 subunit of the PAF complex (*Kelley et al., 1999*; *Krogan et al., 2002*; *Simic et al., 2003*). To aid passage of Pol II, the machinery that travels along with the transcription bubble alters local chromatin structure, yet it is not known how changes to nucleosomes might influence Chd1 or other chromatin remodelers. In addition to potentially generating hexasomes, passage of Pol II is also coupled to transient ubiquitination of H2B (*Fleming et al., 2008*; *Xiao et al., 2005*). Interestingly, the H2B-Ubiquitin (H2B-Ub) mark is required for FACT-assisted disruption of the chromatin barrier (*Pavri et al., 2006*). Chd1 has been shown to be required for high levels of transcription-coupled ubiquitination of H2B in vivo (*Lee et al., 2012*), yet a direct connection between Chd1 and transcriptionally altered nucleosomes has remained elusive.

In this work, we report the discovery that the Widom 601 nucleosome positioning sequence can generate oriented hexasomes, with the sole H2A/H2B positioned in a sequence-defined location. Using oriented hexasomes, we show that Chd1 requires H2A/H2B on the entry side for robust sliding and preferentially shifts hexasomes unidirectionally. Hexasomes can be transformed into nucleosomes upon addition of H2A/H2B dimers, and we demonstrate that oriented hexasomes are an ideal substrate for generating uniform populations of asymmetric nucleosomes with uniquely modified H2A/H2B dimers. We find that nucleosomes with an asymmetric H2B-Ub modification can stimulate nucleosome sliding by Chd1, revealing an unexpected activating role for H2B-Ub in remodeling.

## Results

### The Widom 601 sequence allows for generation of oriented hexasomes

Since the nucleosome consists of two copies each of the four canonical histones – H2A, H2B, H3 and H4 – in vitro nucleosome reconstitutions that deviate from equi-molar histone stoichiometries can result in sub-nucleosomal products. Curiously, during the course of nucleosome reconstitutions by salt dialysis, we noticed that native PAGE migration of a smaller species changed depending on the location of flanking DNA. We use the strong Widom 601 positioning sequence (*Lowary and Widom, 1998*), with the X-601-Y naming convention, where X and Y refer to the number of base pairs

flanking the core 145 bp 601 sequence. Consistently, we observed that the subspecies from 0-601-80 preps migrated faster than that of 80-601-0 preps (*Figure 1A*). We purify nucleosomes, free DNA, and subnucleosomal species away from each other using native polyacrylamide gel electrophoresis (PAGE) (*Figure 1B,C*), allowing for isolation of homogeneous material. The hexasome is a stable sub-nucleosomal particle lacking one of the two H2A/H2B dimers (*Arimura et al., 2012*; *Kireeva et al., 2002*; *Mazurkiewicz et al., 2006*), and we confirmed by SDS-PAGE analysis that the faster migrating species in our nucleosome preparations were in fact hexasomes (*Figure 1D*).

Nucleosomes migrate differently in native gels depending on whether flanking DNA is present only on one or both sides of the histone core (*Eberharter et al., 2004*; *Pennings et al., 1991*). With one H2A/H2B dimer missing, hexasomes have ~40 bp of DNA unwrapped from the core, resulting in DNA lifting off the histone core prematurely at superhelical location 3 (SHL3), three helical turns from the nucleosome dyad, on the side lacking the H2A/H2B dimer. For end-positioned 601 constructs, where nucleosomes lack flanking DNA on one side, hexasomes would be expected to migrate differently depending on whether unwrapping occurred on the side with or without flanking DNA. We reasoned that the differences in hexasome migration may therefore be due to a systematic loss of H2A/H2B from one side of the Widom 601 sequence (*Figure 2A*). To test this idea, we probed the accessibility of DNA using Exonuclease III (ExoIII) (*Figure 2B*). On nucleosomes, ExoIII digestion showed the expected protection at the edge of the histone core, with preferential cleavage in ~10–11 nt increments (lanes 2–4 and 14–16). The 80-601-0 hexasome, in contrast, was digested more internally by ~30–40 nt on the 0 bp side, while showing full nucleosome protection

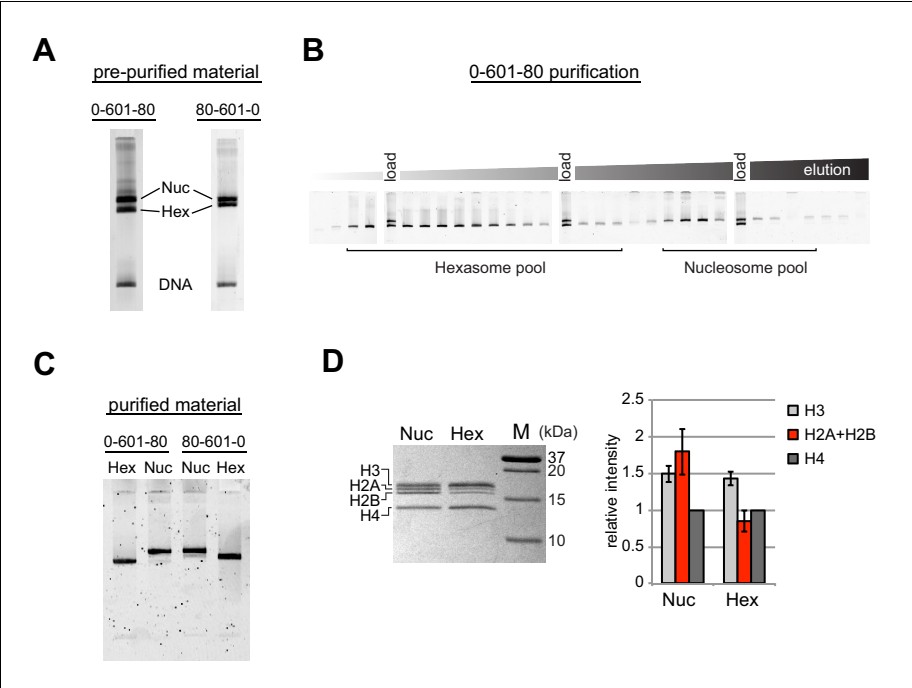

**Figure 1.** Separation of nucleosomes and hexasomes made with the Widom 601 sequence. (**A**) Hexasomes but not nucleosomes migrate differently by native PAGE when flanking DNA is on the left or right of the 601 sequence. These two gels, poured from the same solution, are representative of 0-601-80 and 80-601-0 reconstitutions made using histone octamer. (**B**) Separation of hexasomes from nucleosomes. Shown is a representative purification over a 7% native acrylamide column using a Prep Cell apparatus. The elution fractions were analyzed by native PAGE. (**C**) Purified nucleosome and hexasome pools, analyzed by native PAGE. (**D**) As shown by SDS-PAGE, the hexasome species lack one H2A/H2B dimer. The bar graph is a quantification of gel band intensities from three different nucleosome/hexasome purifications. All histone bands were normalized to histone H4. The H2A and H2B bands often migrate close together, and therefore the relative intensities of H2A/H2B bands are shown summed together. Within each nucleosome/hexasome pair, the intensity of H2A/H2B in hexasomes was 47 ± 6% of that of nucleosomes.

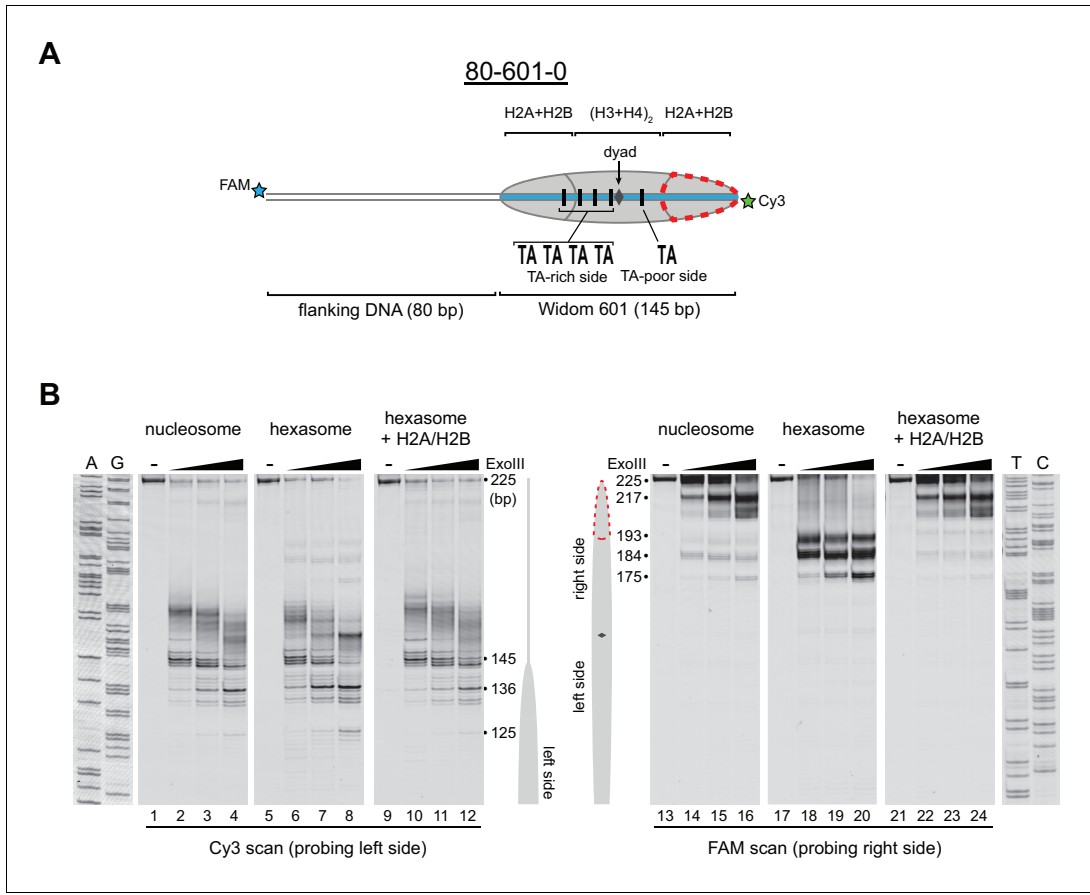

**Figure 2.** Oriented hexasomes can be generated using the Widom 601 sequence. (**A**) Schematic representation of the 80-601-0 nucleosome and hexasome. With limiting amounts of H2A/H2B dimer, the side of the hexasome lacking the dimer (red dotted outline) corresponds with the TA-poor side of the Widom 601 sequence. (**B**) ExoIII analysis of 80-601-0 demonstrates that hexasomes specifically retain the H2A/H2B dimer on the TA-rich side of the 601 sequence. Purified nucleosomes, hexasomes, and hexasomes plus H2A/H2B were incubated with 0, 10, 40, and 160 units of ExoIII and resolved on urea denaturing gels. Lanes 9–12 and 21–24 show addition of 200 nM H2A/H2B dimer to 100 nM hexasomes, which recovered nucleosome digestion patterns. The size (bp) of major products are indicated. These gels are representative of two independent experiments. Dideoxy sequencing lanes (A, G, T, C) were run on the same gel as the samples shown. See also *Figure 2—figure supplements 1* and *2*.

The following figure supplements are available for figure 2:

**Figure supplement 1.** Flanking DNA does not influence the orientation of the hexasome.

**Figure supplement 2.** Sequence and orientation of the Widom 601 sequence used in this study.

on the 80 bp side (lanes 6–8 and 18–20). Relative to the orientation of the 601 sequence, the 0-601-80 hexasome showed an analogous pattern, with ~30–40 nt more extensive ExoIII digestion on the 80 bp side and similar protection on the 0 bp side compared to nucleosomes (*Figure 2—figure supplement 1*). Thus, the preferred location of the remaining H2A/H2B dimer in the hexasome was not influenced by flanking DNA, but instead was determined in a sequence-specific fashion based on the orientation of the Widom 601.

Previous work by several labs has revealed asymmetry in the Widom 601 sequence with respect to the strength of histone-DNA contacts. Single-molecule DNA unzipping experiments demonstrated that one side of the 601 forms more stable contacts with histones (*Hall et al., 2009*), and the asymmetry of the 601 was found to form a polar barrier to passage of RNA polymerase II (*Bondarenko et al., 2006*). One feature that has been pointed out as a key determinant of stable

histone-DNA contacts are periodic TA dinucleotide steps (*Lowary and Widom, 1998*). The Widom 601 is notably asymmetric in TA steps on either side of the dyad where binding affinity is expected to be highest, with four TA steps on one side opposite a single TA step on the other side (*Chua et al., 2012*). Symmetric derivatives of 601 have shown that the TA-rich side is much more salt stable than the TA-poor side (*Chua et al., 2012*), and single molecule experiments have found that the TA-poor side preferentially unwraps under force (*Ngo et al., 2015*). We orient the Widom 601 with the TA-rich side on the left, which means that the side lacking the H2A/H2B dimer in hexasomes corresponds with the TA-poor side of the 601 sequence (*Figure 2A*, *Figure 2—figure supplement 2*).

Others have shown that hexasomes can generate nucleosome-like products upon addition of H2A/H2B dimer (*Kireeva et al., 2002*). To investigate this step-wise method of generating nucleosomes, we incubated hexasomes with a 2-fold molar excess of H2A/H2B dimer and monitored ExoIII digestion. As shown in *Figure 2B*, addition of H2A/H2B dimer yielded a protection pattern indistinguishable from nucleosomes (lanes 22–24). Therefore, even in the absence of histone chaperones or elevated salt, addition of H2A/H2B dimer to hexasomes was sufficient for recovering nucleosome-like protection patterns.

As an alternative method for characterizing hexasomes and nucleosomes generated from H2A/H2B dimer addition, we used histone mapping. With this technique, labeling a single cysteine variant of H2B (S53C) with photo-reactive 4-azidophenacyl bromide (APB) allows for UV-induced cross-linking to nucleosomal DNA (*Kassabov et al., 2002*; *Kassabov and Bartholomew, 2004*). Importantly, cross-linking reduces the chemical stability of the modified base, and therefore favors abasic sites that in turn result in cleavage of the DNA backbone. By separating such site-specifically cleaved fragments on a sequencing gel, the DNA base that reacted with the APB-labeled cysteine can be identified. For chromatin remodelers, changes in positioning of the cross-linked site is interpreted as a shift in the position of the histone core along DNA. In nucleosomes, each H2B cross-links to only one DNA strand, and therefore doubly-labeled fluorescent DNA is needed to report on both sides of the 601 sequence. In agreement with ExoIII experiments, H2B cross-linking for hexasomes was virtually absent on the TA-poor side of the 601, whereas cross-linking on the TA-rich side was equivalent for hexasomes and nucleosomes (*Figure 3*; compare lane 1 with 2 and 4 with 5). Strikingly, addition of H2A/H2B fully recovered the H2B cross-link on the TA-poor side (compare lanes 5 and 6), demonstrating that dimer addition generates correctly organized nucleosomes that are indistinguishable from those obtained by salt dialysis reconstitution. Similar results were obtained with 0-601-80 hexasomes (*Figure 3—figure supplement 1*), reinforcing the conclusion that salt dialysis deposits limiting H2A/H2B on the TA-rich side of the Widom 601.

## Chd1 requires the entry-side H2A/H2B dimer for robust sliding

Given the strong sequence-defined placement of limiting H2A/H2B dimer, we refer to hexasomes produced by the Widom 601 as 'oriented hexasomes'. By having a defined orientation on DNA, these hexasomes offer a unique tool for probing requirements of nucleosome-interacting enzymes. Despite the two-fold pseudo-symmetry of the nucleosome, factors binding off the central dyad axis encounter the two halves of the nucleosome at distinct distances and orientations. The two H2A/H2B dimers and the DNA they coordinate are therefore likely to play unequal roles in nucleosome recognition and enzyme regulation. Chromatin remodelers such as Chd1 shift DNA past the histone core by acting at SHL2, an internal DNA site located ~20 bp from the dyad (*McKnight et al., 2011*; *Saha et al., 2005*; *Schwanbeck et al., 2004*; *Zofall et al., 2006*). Relative to the SHL2 site of DNA translocation, one H2A/H2B dimer is positioned to bind DNA that is pulled onto the nucleosome, and is therefore considered to be on the entry side, whereas the other H2A/H2B binds DNA that shifts off the histone core, and thus is on the exit side. In vitro, Chd1 slides mononucleosomes away from DNA ends (*McKnight et al., 2011*; *Stockdale et al., 2006*). By using end-positioned nucleosomes, we can restrict the direction of sliding, thereby defining H2A/H2B adjacent to the long flanking DNA as the entry-side dimer. Since the placement of the single H2A/H2B dimer relative to 601 is maintained regardless of flanking DNA (*Figure 3* and *Figure 3—figure supplement 1*), we can generate hexasomes with the H2A/H2B dimer on either the entry or exit side, which allows us to determine the extent that Chd1 relies on H2A/H2B at each position.

As a standard technique for visualizing repositioning of mononucleosomes along DNA, we first investigated movement of oriented hexasomes using native PAGE (*Figure 4—figure supplement*

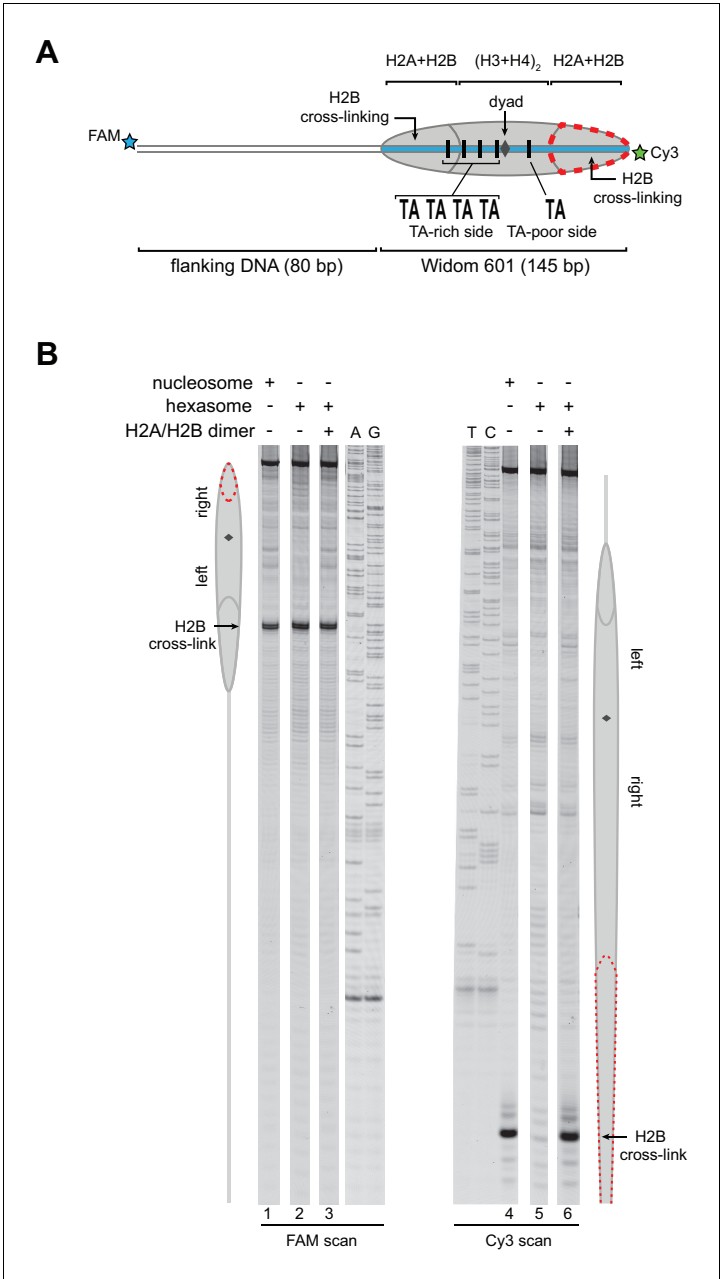

**Figure 3.** Addition of H2A/H2B dimer to hexasomes produces canonical nucleosomes. (**A**) Schematic representation of the 80-601-0 nucleosome and hexasome, highlighting the locations where H2B-S53C cross-links to DNA. Due to the absence of one H2A/H2B dimer, H2B cross-linking with hexasomes is limited to the TA-rich side of the Widom 601. (**B**) Histone mapping demonstrates that canonical nucleosomes can be generated by addition of H2A/H2B dimer to hexasomes. For reactions containing hexasomes plus H2A/H2B, the hexasomes (10 nM) were incubated for 2–3 min with H2A/H2B (20 nM) prior to labeling with APB. Nucleosome and hexasome alone were subjected to the same brief incubation. Following UV cross-linking and DNA extraction, the DNA was cleaved at the crosslinking site and the products separated on a denaturing gel alongside a sequencing ladder to determine the cross-linking position. Results are representative of three or more independent experiments. See also *Figure 3—figure supplement 1*.

The following figure supplement is available for figure 3:

**Figure supplement 1.** Addition of H2A/H2B dimer to hexasomes produces canonical nucleosomes, regardless of flanking DNA location.

*1*). We used 0-601-80 and 80-601-0 constructs described above, which lacked one of the H2A/H2B dimers on either the entry side (0-601-80) or exit side (80-601-0). For nucleosomes, electrophoretic mobility decreased upon addition of Chd1 and ATP, signifying movement away from DNA ends (lanes 1–8). In contrast, under the same conditions the hexasomes failed to show analogous changes in migration patterns (lanes 9–16).

Since the loss of one H2A/H2B dimer in the hexasome dramatically alters the location where DNA extends away from the histone core and thus the geometry of flanking DNA with respect to the core, we considered the possibility that native PAGE may not accurately report on changes in hexasome positioning. We therefore investigated the ability of Chd1 to reposition hexasomes using histone mapping (*Figure 4*). In agreement with previous work (*Patel et al., 2013*; *Stockdale et al., 2006*), Chd1 shifted end-positioned mononucleosomes to more central locations on DNA, with the majority of nucleosomes repositioned ~20 to ~60 bp from their starting locations (lanes 1–3, 7–18). For hexasomes, however, the ability of Chd1 to reposition was strongly dependent on the location of the single H2A/H2B dimer. The 0-601-80 hexasomes, which lacked the entry-side H2A/H2B dimer, failed to show robust repositioning, with the majority of products remaining at the starting position (lanes 4–6). In marked contrast, the 80-601-0 hexasomes shifted robustly onto flanking DNA, demonstrating that Chd1 activity can be supported by an H2A/H2B dimer on only the entry side (lanes 19–30). Interestingly, instead of generating more centrally positioned products, Chd1 shifted 80-601-0 hexasomes to the opposite end of DNA, farther than observed for nucleosomes (*Figure 4B*). This biased movement of 80-601-0 toward the entry H2A/H2B dimer, even after the hexasome had shifted away from its starting position on the 601 sequence, was consistent with much poorer sliding toward the side lacking the H2A/H2B dimer and indicated that it was the absence of H2A/H2B rather than the DNA sequence that blocked efficient sliding of 0-601-80 hexasomes. Thus, Chd1 can reposition hexasomes, but the requirement for entry-side H2A/H2B yields a strong directional bias for hexasomes that contrasts with the back-and-forth sliding observed for nucleosomes.

## Oriented hexasomes allow for precisely designed asymmetric nucleosomes

The discovery of oriented hexasomes opens up a simple means for producing asymmetric nucleosomes, where unique modifications in the two H2A/H2B dimers can be directed to specific sides of the nucleosome. One powerful technique that can benefit from generating asymmetric nucleosomes is single molecule FRET (smFRET). Though many variations are possible, fluorescent dye labeling of nucleosomes commonly involves both histones and DNA, which allows for detection of DNA unwrapping and DNA translocation relative to the histone core (*Blosser et al., 2009*; *Li and Widom, 2004*; *Yang et al., 2006*). The FRET signal, however, can be complicated by the two-fold symmetry of the nucleosome, since dyes at the two related histone positions are typically not equidistant from the DNA-tethered dye, and therefore lead to a mixture of FRET levels (*Deindl et al., 2013*). A standard solution to this issue has been to dilute the labeled histone with an excess of unlabeled histone during nucleosome reconstitution, and select out the desired FRET signal from a single donor/acceptor pair. We expected that the unique placement of a single H2A/H2B dimer relative to the DNA sequence should allow us to generate nucleosomes with a single, uniform FRET pair.

To examine this idea, we labeled H2A-T120C with Cy3-maleimide and generated 3-601-80 hexasomes and nucleosomes containing a DNA-tethered Cy5 dye on the 3 bp side. As previously described (*Deindl et al., 2013*), the nucleosomes gave rise to two major FRET populations corresponding to single Cy3 dyes on the distal or proximal H2A (*Figure 5A*). A mid-FRET population is expected between these two major species, where nucleosomes contain Cy3 on both copies of H2A. Here, due to extensive dilution of labeled H2A-Cy3, that population was minimal. In contrast, the oriented hexasomes yielded a single, high-FRET population as expected for the H2A/H2B dimer located on the exit side, proximal to the DNA label (*Figure 5B*). To see whether these hexasomes would behave as nucleosomes upon H2A/H2B dimer addition, we incubated these samples with Chd1 and ATP to stimulate nucleosome sliding. After a 10 min incubation, all nucleosomes had shifted to a low FRET state, as expected for a ≥20 bp shift of the histone core away from the labeled DNA end. Hexasomes, in contrast, maintained a significant population of high-FRET species after incubation, consistent with the poor movement observed by histone mapping in the absence of entry-side H2A/H2B dimer. Addition of H2A/H2B dimer to hexasomes did not significantly alter the starting high-FRET population, yet incubation with Chd1 and ATP yielded a low-FRET profile similar

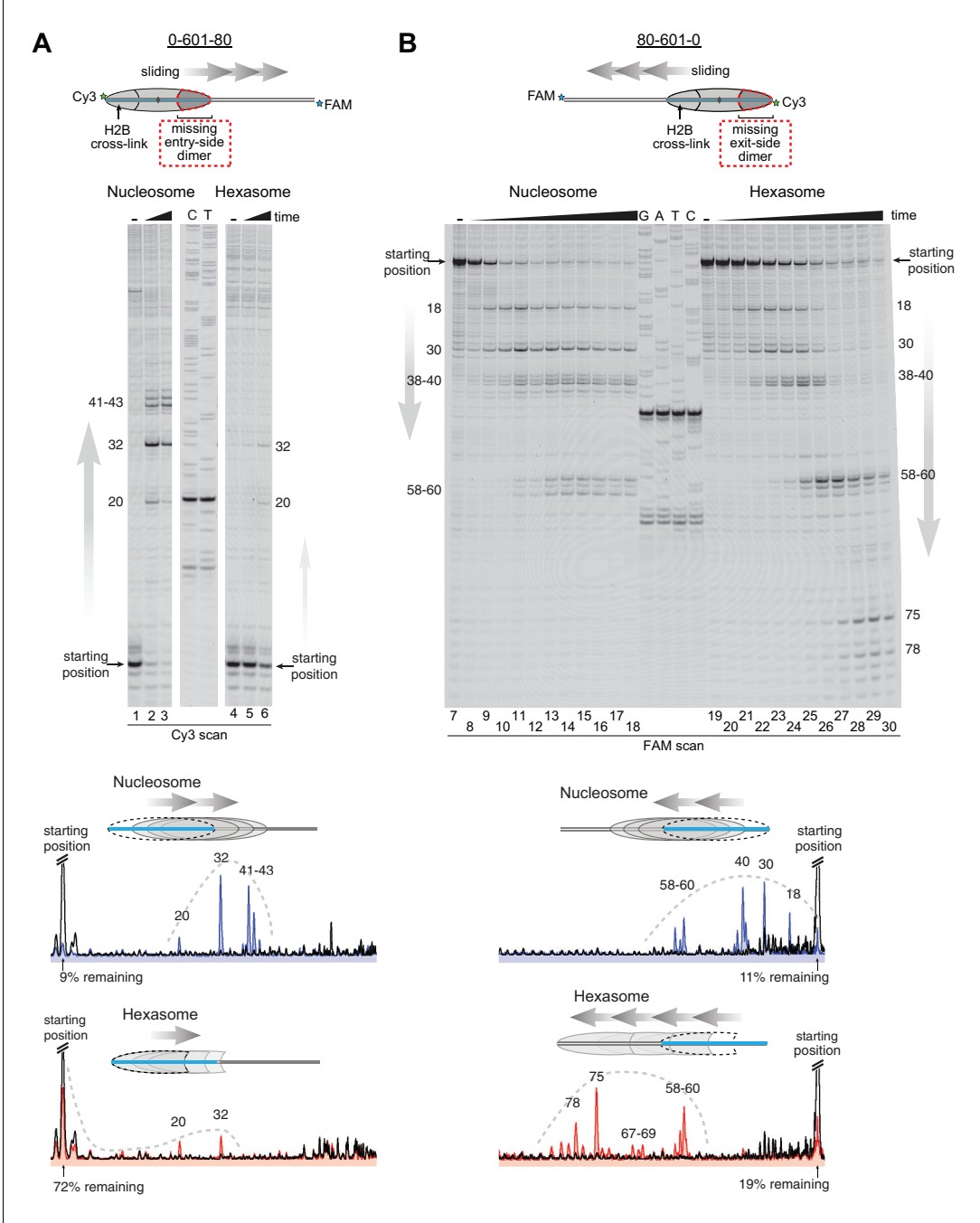

**Figure 4.** Chd1 requires entry side H2A/H2B for robustly repositioning hexasomes. (**A**) Nucleosome and hexasome sliding reactions, visualized through histone mapping. For 150 nM hexasome and nucleosome 0-601-80 constructs, sliding reactions were monitored after incubation with 50 nM Chd1 and 2 mM ATP for 0, 1, and 64 min. Reactions were quenched at time points with the addition of EDTA and competitor DNA. Comparison of intensity profiles for histone mapping reactions are shown below. Samples before ATP addition (0 min) are black, nucleosome sliding reactions after 64 min are blue, and hexasome sliding reactions after 64 min are red. (**B**) Sliding reactions and intensity profiles carried out with 80-601-0 constructs as described for (**A**). Time points were 0, 0.25, 0.5, 1, 2, 4, 8, 16, 32, and 64 min. Sliding experiments for 0-601-80 and 80-601-0 were each performed six or more times with similar results. See also ***Figure 4—figure supplement 1***.

The following figure supplement is available for figure 4:

*Figure 4 continued on next page*

*Figure 4 continued*

**Figure supplement 1.** Chd1 remodeling dramatically alters nucleosome but not hexasome mobility as assessed by native PAGE.

to that observed for nucleosomes (*Figure 5C*). These results show that oriented hexasomes offer a defined methodology for producing uniformly labeled nucleosomes that should benefit smFRET experiments.

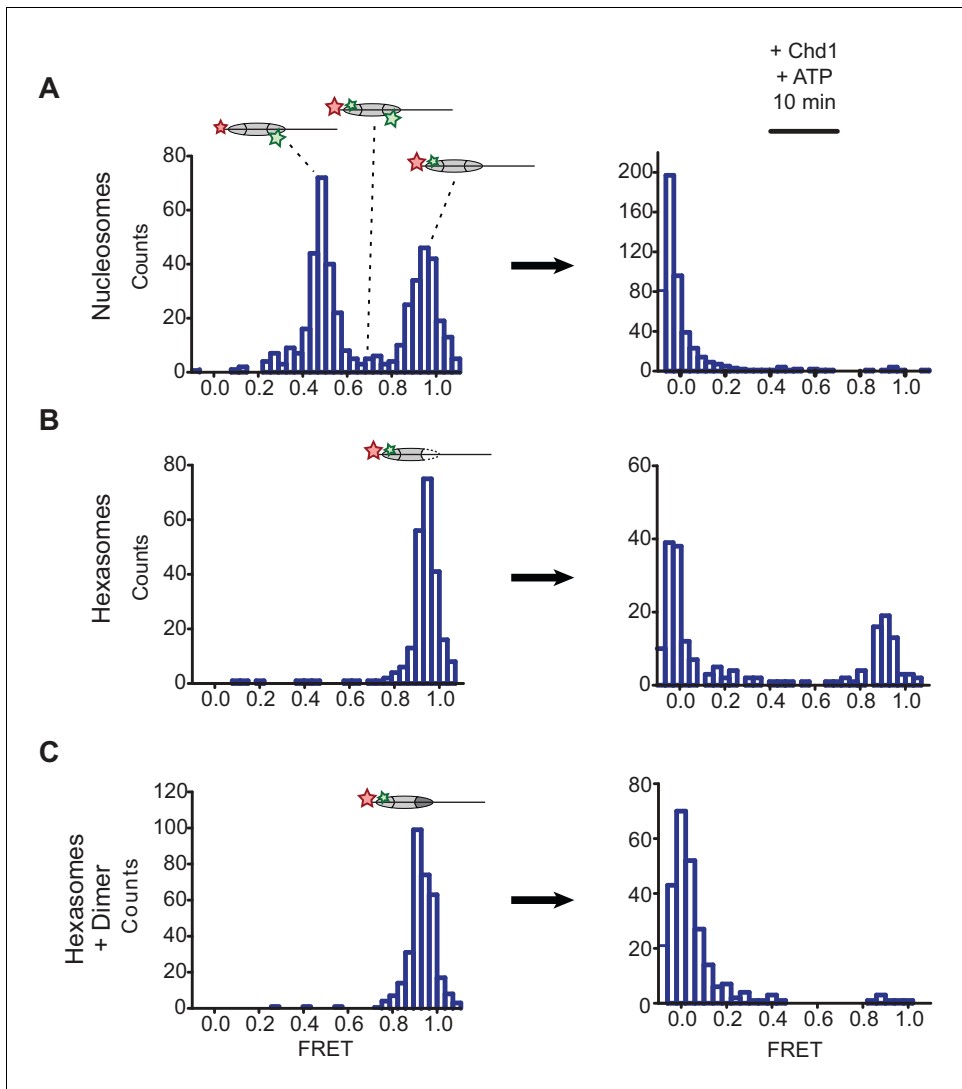

**Figure 5.** Oriented hexasomes allow targeted placement of modified H2A/H2B dimers on the nucleosome. (**A**) Analysis of dual labeled 3-601-80 nucleosomes (H2A T120C-Cy3 and DNA-Cy5) by single-molecule FRET (smFRET) reveals multiple species prior to nucleosome sliding by Chd1. Nucleosomes were surface-immobilized by biotin on the 80 bp flanking DNA. Infusion of 300 nM Chd1 and ATP initiated remodeling. (**B**) Oriented 3-601-80 hexasomes (H2A-Cy3 and DNA-Cy5) uniformly show one dye pair that yields high FRET. Right panel shows relatively poor mobilization of hexasomes by Chd1. (**C**) Incubation of a two-fold molar excess of unlabeled H2A/H2B dimer with the labeled 3-601-80 hexasomes yielded asymmetric nucleosomes, only possessing the high FRET dye pair. After remodeling with Chd1 and ATP, the FRET population was similar to nucleosome.

## Chd1 requires entry side H2A/H2B for sliding but not binding

From the experiments presented above, it was unclear whether the asymmetric sliding of hexasomes simply reflected much poorer binding of Chd1 to the side lacking the H2A/H2B dimer. To see whether the remodeler engaged with hexasomes differently than nucleosomes, we utilized single cysteine variants of Chd1 that allow for site-specific cross-linking to nucleosomal DNA. In the presence of the ATP analog ADP•BeF$_3$, both lobes of the ATPase motor of Chd1 bind to DNA at SHL2, which can be monitored by APB labeling of the Chd1 variants N459C (lobe 1) and V721C (lobe 2) (*Nodelman et al., 2017*). Using 300 nM of either Chd1 variant with 150 nM 40-601-40 nucleosomes, cross-linking was observed to SHL2, 15 to 19 bp from the dyad on either side of the nucleosome as expected (*Figure 6A* – lanes 2, 5, 8 and 11). Strikingly, the same cross-linking pattern was also observed with 40-601-40 hexasomes (lanes 3, 6, 9 and 12), indicating that this ATP-bound state of the ATPase motor was not hindered by the lack of H2A/H2B on one side. These results suggest that the deficiency in sliding hexasomes lacking entry side H2A/H2B was likely a catalytic rather than a binding defect.

To quantitatively analyze the impact of entry side H2A/H2B dimer on Chd1 sliding, we monitored repositioning of the histone core using a real-time assay based on static quenching of fluorescence (SQOF). Using the same labeling positions described above for FRET, we found that quenching can be achieved using either a donor (Cy3B) and quencher (Dabcyl) pair, or two cyanine dyes (Cy3-Cy3). As with FRET, movement of exit DNA away from the histone core separates donor-quencher pair and results in increased donor fluorescence. Since quenching requires direct contact, SQOF likely provides higher sensitivity than FRET at shorter distances. With Dabcyl as a quencher on exit DNA, we monitored fluorescence of 0-601-80 hexasomes containing a single H2A-Cy3B label in the absence and presence of an additional, unlabeled H2A/H2B dimer. Reactions were performed with saturating (1 mM) ATP and excess Chd1 (600 nM) relative to hexasome (10 nM) to reduce the likelihood that defects in sliding might be attributed to binding. Under these conditions, Chd1 shifted the hexasome much more rapidly when excess H2A/H2B was added to generate nucleosomes (*Figure 6B*). Both reactions were fit to double exponentials, with hexasome alone dominated by a slower rate of 0.0017 s$^{-1}$, whereas hexasome plus H2A/H2B dimer yielded a dominant, >300 fold faster rate of 0.57 s$^{-1}$. The hexasome reactions also yielded a ~three-fold lower change in fluorescence intensity compared to nucleosomes, consistent with an inability to shift all hexasomes away from the DNA end under equilibrium conditions and indicating that even the slow hexasome rate is likely an overestimate. Taken together, these experiments demonstrate that the ability of Chd1 to shift hexasomes lacking the entry side H2A/H2B dimer is extremely poor.

We also compared sliding reactions for hexasome plus H2A/H2B dimer versus salt-dialyzed nucleosomes. As expected, these two substrates yielded similar progress curves, though hexasome plus dimer was slightly faster. This difference arose from a modest (~15%) decrease in the fast rate (0.49 s$^{-1}$ vs 0.57 s$^{-1}$) and a larger contribution of the slow rate to the fits (11% vs 1%) for salt dialyzed nucleosome compared with hexasome plus H2A/H2B dimer. Previous kinetic analysis of nucleosome sliding by the ACF remodeler using FRET reported that a ~10 fold slower phase contributed 10% of the signal, which the authors suggested was due to a distinct population of nucleosome substrates (*Yang et al., 2006*). This explanation matches the behavior we observed for nucleosomes here, and suggests that the faster shifting population of nucleosomes is more favored with H2A/H2B dimer addition to hexasomes. We also repeated comparison of nucleosomes versus hexasome plus dimer substrates at a lower (25 µM) ATP concentration (*Figure 6—figure supplement 1*). Under these conditions, we found no significant difference in the fast rate of sliding, again with the slow rate contributing more to the amplitude of the progress curve for nucleosome (10%) compared to hexasome plus H2A/H2B (3%). These results suggest that adding H2A/H2B to hexasomes produces nucleosomes that are similar to and apparently more homogeneous than those produced by salt dialysis.

## The H2A acidic patch is not essential for nucleosome sliding by Chd1

One possible explanation for the sliding defect of hexasomes is that Chd1 makes a critical contact with the entry side H2A/H2B dimer. To see if we could identify an important epitope required for sliding, we generated five H2A/H2B variants that could be used to transform hexasomes containing one wild type H2A/H2B into a nucleosome with an altered H2A/H2B on the entry side. Since each of the H2A/H2B variants were generated in distinct preparations, however, we were concerned about

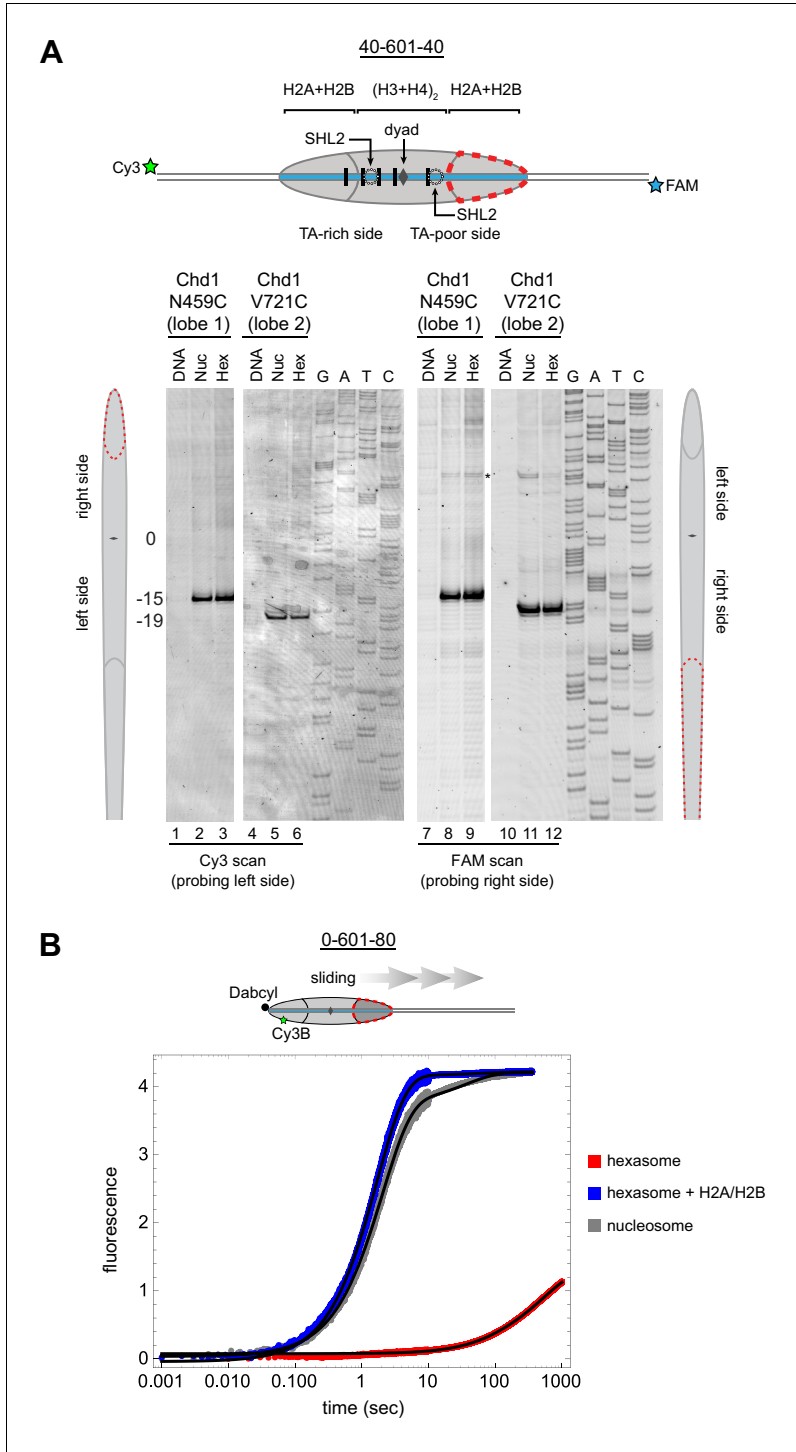

**Figure 6.** Chd1 requires entry side H2A/H2B for sliding but not binding. (**A**) Chd1 cross-linking to 40-601-40 nucleosomes and hexasomes. Single cysteine variants on lobe 1 (N459C) and lobe 2 (V721C) of the Chd1 ATPase cross-linked to DNA 15 and 19 bp from the dyad, respectively, on both sides of nucleosomes and hexasomes. Chd1 was labeled with APB and incubated in a 2:1 ratio with DNA, nucleosomes, or hexasomes in the presence of ADP•BeF₃. After UV irradiation, DNA extraction and cleavage, cross-linking sites were determined by separating DNA fragments on a denaturing gel alongside a sequencing ladder. The gel shown is representative of two independent experiments. Asterisk marks cross-linking from a non-cysteine residue (*Nodelman et al., 2017*). (**B**) Stopped flow sliding reactions comparing the activity of Chd1 on 10 nM 0-601-80 nucleosomes, hexasomes, and hexasomes plus 12 nM dimer. Nucleosomes and hexasomes were labeled with Cy3B on H2A-T120C and with

*Figure 6 continued on next page*

*Figure 6 continued*

Dabcyl quencher on the zero end of the DNA. Reactions were initiated with the addition of saturating (600 nM) Chd1 and 1 mM ATP. Black lines represent double exponential fits of the data. Each progress curve is an average of 3–6 replicate injections, and representative of two independent experiments. See also *Figure 6—figure supplement 1*.

The following figure supplement is available for figure 6:

**Figure supplement 1.** Chd1 repositions nucleosome and hexasome plus dimer at similar rates with limiting ATP.

two possible complications associated with H2A/H2B dimer addition: too little dimer would allow a significant fraction of hexasome to remain in the reaction, which might compete with the nucleosome, whereas too much dimer could create off-products that might interfere with nucleosome sliding.

To evaluate the potential effects of dimer concentration on sliding, we used Cy3-Cy3 SQOF to monitor sliding of 0-601-80 hexasomes in the presence of increasing amounts of wild type H2A/H2B dimer. Relative to the constant hexasome concentration used (10 nM), addition of unlabeled H2A/H2B dimer stimulated sliding at all concentrations, from undersaturating (4 nM) to saturating (24 nM) (*Figure 7—figure supplement 1A*). The total change in fluorescence intensity increased with dimer concentration, consistent with only the fraction of hexasomes converted to nucleosomes being readily shifted by Chd1. A maximum change in fluorescence was observed with a 1.6-fold molar ratio of dimer to hexasome (*Figure 7—figure supplement 1B*), which is in agreement with others who have reported requiring ~2 fold molar equivalent of dimer to convert all hexasome to nucleosome (*Kireeva et al., 2002*).

Interestingly, the reaction rates were maximal and remained constant over a wide range of H2A/H2B concentrations, from subsaturating up to the 1.6-fold molar equivalent that yielded the maximum change in fluorescence intensity (*Figure 7—figure supplement 1C*). Thus, under the conditions used here, the presence of hexasome due to limited H2A/H2B dimer addition did not influence rates for Chd1 remodeling. Beyond this saturating amount, both the rates and range of fluorescence intensity decreased. These reductions likely resulted from improper H2A/H2B deposition on flanking DNA that interfered with Chd1 action. We also monitored nucleosome formation by native PAGE, which showed a dimer-mediated shift of the hexasome species to nucleosomes and aggregation with excessive H2A/H2B (*Figure 7—figure supplement 1D*). As the most consistent rates were observed below the two-fold molar equivalent, we used only a slight molar excess of dimer for the remainder of our dimer addition experiments (10 nM hexasome plus 12 nM H2A/H2B).

In an attempt to identify a critical epitope required for Chd1 sliding, we introduced site-specific disruptions in three locations of the H2A/H2B dimer: (1) a potential binding surface on H2B; (2) the C-terminus of H2A; and (3) the H2A acidic patch (*Figure 7A*). We used Cy3B-Dabcyl SQOF to compare the activity of Chd1 on nucleosomes containing disruptions at these sites on the entry side H2A/H2B.

Histone H2B possesses a conserved hydrophobic patch that was recently shown to be recognized independently by both Spt16 and Pob3 subunits of FACT (*Kemble et al., 2015*). Interestingly, the residues bound by FACT (Y45 and M62 in yeast) are also recognized by the catalytic subunit of the SWR1 remodeler (*Hong et al., 2014*), highlighting the importance of this patch in H2A/H2B dimer recognition. To see if this region is also important for Chd1, we substituted both H2B residues (Y39 and M56 in *Xenopus*) with alanine. Nucleosome sliding rates with this H2B variant (*Figure 7*, cyan) were indistinguishable from wild type, indicating that Chd1 does not require these residues on the entry side dimer for normal activity.

Another epitope on H2A/H2B that has been shown to be important for a chromatin remodeler is the H2A C-terminal tail. In previous work, nucleosomes lacking H2A C-terminal residues 115–129 were repositioned more poorly by ISWI remodelers, even though this C-terminal deletion was reported to facilitate heat shifting of histone octamers (*Vogler et al., 2010*). Since the H2A C-terminus was reported to not be required for octamer stability (*Bao et al., 2004*), we used a truncated H2A lacking residues 110–129. Interestingly, Chd1 mobilized nucleosomes containing this truncated H2A on the entry dimer 2.5-fold faster than wild type (*Figure 7*, green). Thus, the H2A C-terminus

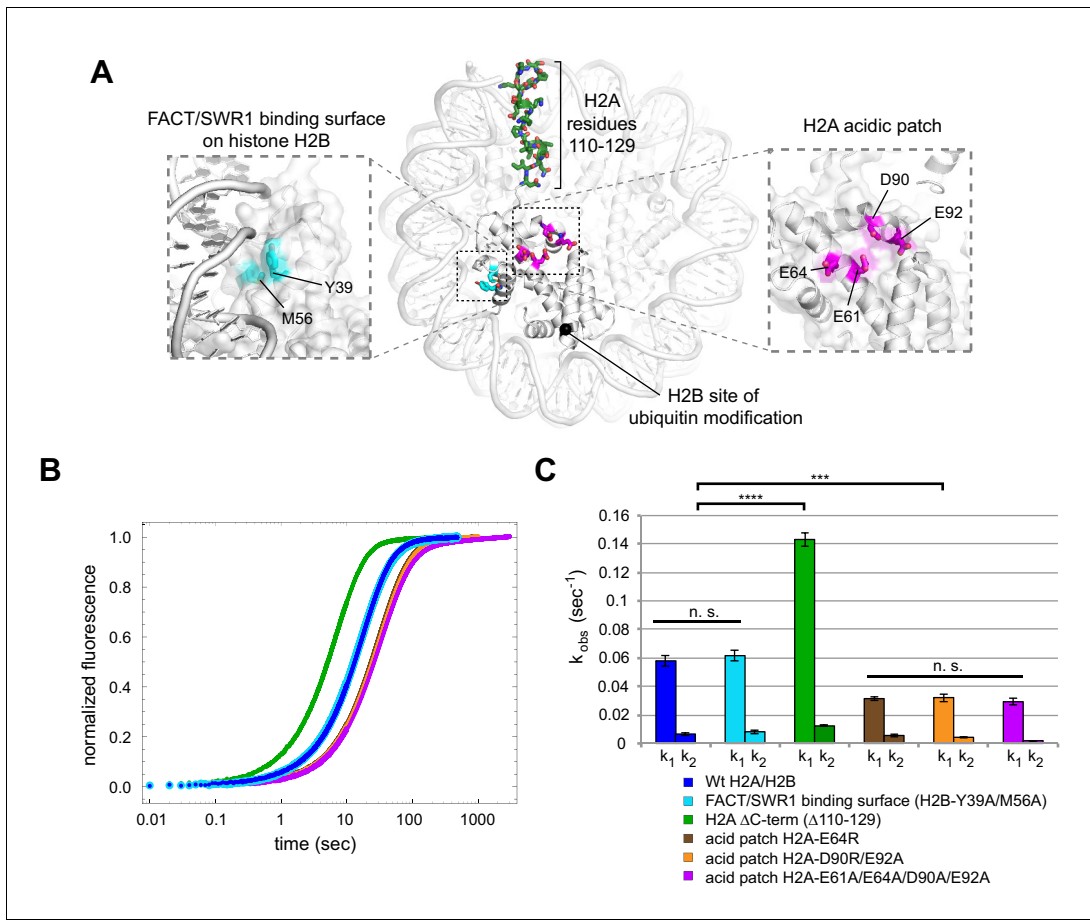

**Figure 7.** Disruptions in the nucleosome acid patch only moderately decrease sliding by Chd1. (**A**) Overview of disruptions introduced on H2A/H2B. Nucleosome crystal structure shown is PDB code 1KX5 (*Davey et al., 2002*). (**B**) Stopped flow sliding reactions using asymmetric nucleosomes containing H2A or H2B disruptions on the entry side H2A or H2B. Asymmetric nucleosomes were generated by incubating 10 nM 0-601-80 hexasomes with 12 nM H2A/H2B containing one of the following sequence variants: Wt (blue); FACT/SWR1 binding surface disruption (H2B-Y39A/M56A) (cyan); H2A C-terminal tail truncation (Δ110–129) (green); acid patch single mutant (H2A-E64R) (brown); acid patch double mutant (H2A-D90R/E92A) (orange); acid patch quadruple mutant (H2A-E61A/E64A/ D90A/E92A) (magenta). Reactions were performed with 400 nM Chd1 and 25 μM ATP and followed by Cy3B-Dabcyl SQOF. Each progress curve is an average of 3–6 technical replicates. (**C**) Summary of observed rates ($k_1$, $k_2$) obtained from double exponential fits to stopped flow data as shown in (**A**). In every case the observed fast rate ($k_1$) contributes >90% of the amplitude of the progress curve. Error bars represent standard deviation from three (six for Wt) independent experiments. Statistics compare $k_1$ rates for indicated constructs: *** p-value <0.00001; **** p-value <0.0000001; n.s., not significant. See also *Figure 7—figure supplements 1* and *2*.

The following figure supplements are available for figure 7:

**Figure supplement 1.** With subsaturating H2A/H2B dimer addition, rates of nucleosome sliding by Chd1 are not sensitive to nucleosome:hexasome ratios.

**Figure supplement 2.** With limiting ATP, remodeling saturates at 400 nM Chd1.

does not contain a critical epitope for Chd1, and faster sliding may have been achieved by altered histone-histone or histone-DNA dynamics.

Finally, the H2A acidic patch, which differs among H2A variants, has been found to be a critical epitope for several nucleosome-interacting factors (*Kalashnikova et al., 2013*). In fact, direct interactions with the H2A acidic patch occur in all of the nucleosome co-crystal structures solved to date: the LANA peptide from Kaposi's sarcoma-associated herpesvirus (*Barbera et al., 2006*), the Sir3

BAH domain (*Armache et al., 2011*), RCC1 (*Makde et al., 2010*), the ubiquitylation module of PRC1 (*McGinty et al., 2014*), and the SAGA deubiquitinating module (*Morgan et al., 2016*). We introduced three combinations of amino acid substitutions of the H2A acidic patch: H2A-E64R, H2A-D90R/E92R, and H2A-E61A/E64A/D90A/E92A. In comparison to wild type H2A, each of these H2A mutations resulted in a two-fold decrease in Chd1 nucleosome sliding rates (*Figure 7*, brown, orange, and magenta). These modest rate decreases suggest that the H2A acidic patch is not a critical epitope required for Chd1 sliding. However, it is interesting to note that these reactions were performed with saturating Chd1 (*Figure 7—figure supplement 2*), and therefore the reduced rate, while modest, suggests a catalytic rather than a binding defect.

## Chd1 is specifically stimulated by ubiquitinated H2B on the entry-side dimer

As an alternative to disrupting a surface of H2A/H2B, we explored the addition of a ubiquitin modification to H2B. We reasoned that the large ubiquitin moiety may block access of Chd1 to a critical dimer epitope required for robust Chd1 sliding. Additionally, based on the close ties of both Chd1 and H2B ubiquitination with transcribing Pol II (*Kelley et al., 1999*; *Krogan et al., 2002*; *Simic et al., 2003*; *Xiao et al., 2005*), Chd1 likely encounters ubiquitinated nucleosomes, and it would therefore be biologically relevant to understand the impact of H2B-ubiquitination on Chd1 activity.

We used oriented hexasomes to produce nucleosomes with the four combinations of modified and unmodified H2A/H2B dimers. To assemble nucleosomes with different placements of ubiquitin, we first generated 0-601-80 hexasomes with the exit dimer either unlabeled (Wt) or chemically cross-linked to ubiquitin (Ub) via a cysteine introduced at the H2B C-terminus (*Long et al., 2014*). To each of these hexasomes, H2A/H2B dimer (Wt or Ub) was then added to produce the four combinations of Wt/Ub nucleosomes. Since ubiquitin significantly alters the sizes and shapes of hexasomes and nucleosomes, analysis by native gel clearly demonstrated that each reaction possessed a unique nucleosome with the desired placement of modified and unmodified H2A/H2B dimers (*Figure 8A*).

With these four nucleosome species, sliding reactions monitored by Cy3-Cy3 SQOF were carried out to determine whether H2B-Ub affected Chd1 activities. Despite the significant size of ubiquitin and potential to block access to H2A/H2B, the presence of this modification did not impede nucleosome sliding by Chd1. In fact, the two nucleosomes containing entry-side H2B-Ub (Wt-Ub and Ub-Ub) yielded faster rates than nucleosomes with unmodified entry-side H2B (*Figure 8B*). These results reinforce the finding that Chd1 activity is sensitive to the entry-side dimer, and reveal that Chd1 is stimulated by H2B-Ub.

We used the Cy3B-Dabcyl pair to measure the rates of Chd1 sliding for nucleosomes with and without H2B-Ub on the entry side, generated from oriented hexasomes. In agreement with a preferential stimulation of Chd1, nucleosomes containing entry-side H2B-Ub consistently showed faster rates of sliding (*Figure 8C,D*). These experiments were conducted with saturating Chd1 (400 nM) indicating that H2B-Ub did not merely improve Chd1 binding but increased catalytic turnover. The presence of the ubiquitin moiety could accomplish this either by helping to retain Chd1 on the nucleosome thereby increasing processivity/productivity from one enzyme-binding event, or by predisposing Chd1 to adopt an active conformation on the nucleosome.

## Discussion

Here we show that the Widom 601 can be used to make oriented hexasomes, a unique and powerful tool for studying chromatin-interacting factors. We generated hexasomes from nucleosome reconstitutions in the presence of limiting H2A/H2B dimer, and our work indicates that one side of the 601 has a much higher affinity for H2A/H2B than the other, resulting in preferential salt-deposition of the dimer in a sequence-specific fashion (*Figure 2*). While identification of sequence elements responsible for preferred H2A/H2B deposition requires further investigations, we speculate that the biased orientation of hexasomes may arise from the asymmetric distribution of inward-facing minor groove TA steps flanking the nucleosome dyad. The Widom 601 is well known for its marked asymmetry in strength of histone-DNA contacts (*Hall et al., 2009*), and higher salt resistance of the TA-rich side of the 601 (*Chua et al., 2012*) is consistent with the preferential H2A/H2B deposition that yields oriented hexasomes. Although primarily associated with H3/H4 on the most internal portion of

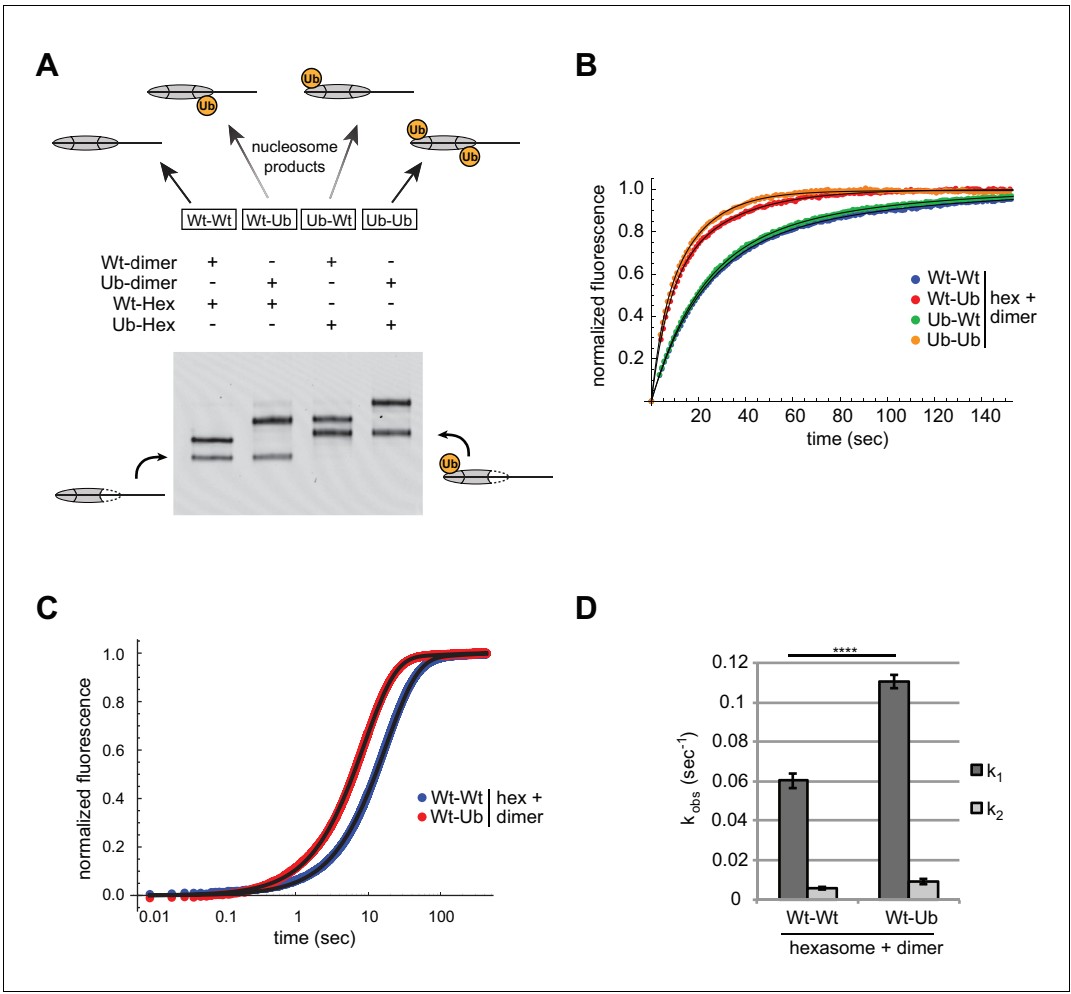

**Figure 8.** Entry-side H2B-Ubiquitin stimulates nucleosome sliding by Chd1. (**A**) Generation of symmetric and asymmetric nucleosomes with site-specific placement of H2B-Ubiquitin. Nucleosomes were formed from subsaturating H2A/H2B dimer (12 nM) addition to 0-601-80 hexasomes (10 nM). Hexasomes and H2A/H2B dimer contained either unmodified (Wt) or ubiquitinated (Ub) H2B as indicated, and resulting nucleosome and hexasome species were visualized by native PAGE. Shown is a representative from six independent dimer addition experiments. (**B**) Comparison of remodeling reactions with subsaturating (25 nM) Chd1, using hexasomes (10 nM) and H2A/H2B dimers (12 nM) containing unmodified or Ub-conjugated H2B. Shown are progress curves for remodeling reactions monitored using a Cy3-Cy3 pair at 25 μM ATP. Black traces represent fits to the data. Progress curves are representative of two independent experiments. (**C**) Representative progress curves of nucleosome sliding reactions monitored by stopped flow using Cy3B-Dabcyl at 25 μM ATP and saturating (400 nM) Chd1. Each progress curve is an average of 3–6 technical replicates. Black traces represent fits to the data. (**D**) Comparison of observed sliding rates monitored with Cy3B-Dabcyl at 25 μM ATP and saturating Chd1 (400 nM). Error bars show standard deviations from three independent experiments. **** p-value <0.0001.

nucleosomal DNA, the periodic TA steps also correlate with interactions between H2A/H2B and the adjacent segments of DNA. In force pulling experiments, the periodic TA steps were shown to influence unwrapping, with a strong preference for unwrapping the TA-poor side of the Widom 601 (**Ngo et al., 2015**). The sequence and structural properties of the DNA segment directly contacting H2A/H2B should be important for dimer affinity, and notably, one of the four TA-step positions is located at SHL3.5, a minor groove site contacted directly by H2A/H2B dimer, which is 'TA' on the TA-rich side and 'CC' on the TA-poor side (**Figure 2—figure supplement 2**). While DNA that directly contacts H2A/H2B likely plays a role, we believe that the stability of the adjacent H3/H4-DNA interactions, dictated by the presence or absence of periodic TA steps surrounding the dyad

may be just as important in determining H2A/H2B occupancy. The strength of histone-DNA interactions also depends on the nature of the histones themselves. Here, we focused on the widely used canonical histones from *Xenopus laevis*, and it will be interesting to discover the extent that histone variants and canonical histones of other species respond to DNA sequence elements of the Widom 601 and in naturally occurring nucleosome positioning sequences.

For chromatin remodelers, each H2A/H2B dimer is in a unique position, engaging with DNA either entering or exiting the nucleosome. Like other remodelers, the ATPase motor of Chd1 shifts DNA toward the dyad (*McKnight et al., 2011*), which means that the SHL2 site where the motor acts is adjacent to the entry-side H2A/H2B dimer. We show that the entry-side dimer is critical for robust sliding by Chd1, which results in a strongly biased repositioning of hexasomes toward the side with the H2A/H2B dimer (*Figure 4*). As shown by cross-linking, the absence of one H2A/H2B dimer does not appear to diminish binding of the Chd1 ATPase motor to either SHL2 site of the hexasome (*Figure 6A*), strongly suggesting that the defect in sliding occurs after engagement of the remodeler.

While disruption of the H2A acidic patch modestly decreased Chd1 sliding, we were unable to identify an epitope on the entry side H2A or H2B that mimicked the dramatic loss of sliding activity seen with hexasomes. One explanation for our findings is that rather than a specific interaction with the entry side H2A/H2B dimer, Chd1 may instead be responding to DNA unwrapping. How Chd1 might sense unwrapping on the entry side is not yet clear, but loss of DNA wrapping from an H2A/H2B dimer may alter dynamics of histone-histone and the remaining histone-DNA contacts, providing a means for Chd1 to indirectly determine the state of the nucleosome prior to sliding. Sensitivity to DNA unwrapping is consistent with slower nucleosome sliding activity of Chd1 when a transcription factor is bound on the entry side of the nucleosome (*Nodelman et al., 2016*). Transcription factors compete with histone-DNA contacts and can dramatically unwrap nucleosomal DNA when their binding sites are located within the histone footprint (*Li and Widom, 2004*; *North et al., 2012*). By slowing nucleosome sliding by Chd1, analogously to what we observe here with hexasomes, DNA unwrapping could provide a means for sensing a transcription factor at the nucleosome edge, which in turn would help avoid pulling transcription factors further onto the nucleosome. Although experiments performed here were limited to the Chd1 remodeler, we expect that the biochemically similar ISWI remodelers, which slide nucleosomes directionally away from bound transcription factors and generate evenly spaced nucleosome arrays (*Kang et al., 2002*; *Li et al., 2015*; *Lusser et al., 2005*), should also exhibit a strong directional bias in sliding hexasomes.

The directional sliding of hexasomes by Chd1, which contrasts with the back-and-forth movement typical for nucleosomes, likely influences chromatin organization in vivo. In vitro, both transcription through nucleosomes by Pol II and remodeling by RSC along with the NAP1 chaperone have been shown to generate hexasomes (*Kireeva et al., 2002*). Although histone chaperones such as FACT would be expected to replace the missing H2A/H2B dimer during transcription, the passage of Pol II has been shown to specifically displace the H2A/H2B dimer distal to the promoter (*Hsieh et al., 2013*; *Kulaeva et al., 2009*), which would orient hexasomes for biased sliding toward the 5' end. One speculative idea is that nucleosome array packing against the +1 nucleosome, which is dependent on Chd1 and ISWI chromatin remodelers (*Gkikopoulos et al., 2011*; *Zhang et al., 2011*), would be favored by directional hexasome sliding (*Figure 9*). Even with relatively few or transient hexasomes, directional sliding toward the transcriptional start site would be expected to corral upstream nucleosomes, similar to the phasing of nucleosome arrays against transcription factor-targeted nucleosomes (*McKnight et al., 2016*; *Wiechens et al., 2016*). Consistent with this idea, it has been shown that inactivation of Pol II relaxes nucleosome packing in coding regions, resulting in a nucleosome drift of ~10 bp toward the 3' end of yeast genes (*Weiner et al., 2010*).

Our work also demonstrates how hexasomes made using the Widom 601 are a useful tool for generating specifically oriented asymmetric nucleosomes. To extract meaningful information on nucleosome dynamics, single molecule FRET experiments require a single, specifically positioned donor-acceptor pair (*Blosser et al., 2009*; *Deindl et al., 2013*; *Ngo et al., 2015*). However, due to the pseudo two-fold symmetry of the nucleosome, labeling any of the histones with a FRET fluorophore (donor or acceptor) typically yields three different labeling configurations (*Deindl et al., 2013*). The presence of multiple FRET pairs, each with distinct and potentially closely spaced FRET values substantially complicates such analyses and at the same time limits throughput by decreasing the population of nucleosomes with the desired labeling configuration. Here, we show how oriented

**Figure 9.** Model for nucleosome packing by oriented hexasomes. As others have shown, transcription by Pol II through nucleosomes is facilitated by removal of the promoter-distal H2A/H2B dimer (*Kulaeva et al., 2009*). Our results indicate that Chd1 would slide a hexasome of this orientation upstream. We propose that one or more hexasomes would corral intervening nucleosomes toward the promoter. Alternately, if every transcribed nucleosome were briefly converted to a hexasome, unidirectionally sliding of each hexasome would maintain tight nucleosome packing.

hexasomes can yield a homogeneously labeled population (*Figure 5*) that greatly facilitates smFRET experiments.

We furthermore demonstrate the utility of oriented hexasomes for generating asymmetric nucleosomes containing a uniquely positioned H2A/H2B PTM (*Figure 8A*). This method complements a recent procedure described for generating nucleosomes with distinct H3 tails (*Lechner et al., 2016*), and has the added advantage that the standard nucleosome reconstitution is sufficient for making oriented hexasomes, which can then be readily transformed into asymmetric nucleosomes with H2A/H2B dimer addition. Given the prevalence of asymmetric histone modifications and variants, this methodology should aid further investigation into how spatial cues contribute to the histone code, especially when used in conjunction with recent advances in detecting modifications on asymmetric nucleosome substrates (*Liokatis et al., 2016*).

Using this methodology, we demonstrate that ubiquitin-modified H2B stimulates Chd1, but only when present on the entry-side dimer (*Figure 8*). Unexpectedly, the stimulatory effect of ubiquitin was observed even with saturating remodeler concentrations, indicating that it is not merely the result of improved Chd1 binding. While we favor the idea of a direct interaction, where ubiquitin stabilizes an active conformation of Chd1 on the nucleosome, it is also possible that H2B-Ub alters the structure or dynamics of the nucleosome itself. Interestingly, H2B-Ub is required for maximum stimulation of Pol II transcription by FACT, potentially through aiding displacement of H2A/H2B (*Pavri et al., 2006*).

The stimulatory effect of ubiquitin raises new questions regarding how histone modifications bias chromatin remodelers, and more broadly, the discovery of oriented hexasomes should be a valuable tool for deepening our understanding of chromatin biology.

## Materials and methods

### Protein production and modifications

Expression and purification of proteins used in this study were carried out as previously described for a truncated form of *Saccharomyces cerevisiae* Chd1 (residues 118–1274) (*Hauk et al., 2010*; *Patel et al., 2011*), *Xenopus laevis* histones (*Luger et al., 1999*), and the ubiquitin variant G76C (*Long et al., 2014*). The ubiquitin sequence used in this work was identical to human and *X. laevis*, which is 96% identical to *S. cerevisiae* ubiquitin.

Conjugation of H2B and ubiquitin (H2B-Ub) was carried out essentially as described (*Long et al., 2014*) to produce a nonhydrolyzable H2B-Ub mimic. A cysteine was introduced in place of the C-terminal lysine of *X. laevis* H2B (annotated as K117 in crystal structures using *X. laevis* histones, and equivalent to K120 in mammalian and full length *X. laevis* H2B). Based on the concentration of

reduced cysteines using Ellman's reagent, H2B (K117C) and His-tagged Ubiquitin (G76C) were combined at a 2:1 ratio at a protein concentration of ~10 mg/mL in denaturing conditions. TCEP (5 mM $C_f$) was added and incubated for 30 min. Crosslinking of the two proteins was carried out by adding 100 mM 1,3 dichloroacetone to a final amount equal to half the total moles of reduced cysteines, and then quenching with 2-mercaptoethanol after 45 min. Un-crosslinked histones were removed using nickel affinity purification under denaturing conditions. The amount of crosslinked H2B-Ub was estimated from SDS-PAGE and refolded at a 1:1 ratio with *X. laevis* H2A. The H2A/H2B-Ub dimer was purified by size exclusion chromatography as described for unmodified H2A/H2B (*Dyer et al., 2004*). For fluorescently tagged histones, H2A(T120C) was labeled with maleimide derivatives of Cy3 or Cy3B prior to refolding as previously described (*Shahian and Narlikar, 2012*).

## Production of hexasomes and nucleosomes

*Xenopus laevis* histones were refolded in equimolar ratios to obtain dimers (H2A/H2B), tetramers (H3/H4)$_2$, and octamers (H3/H4/H2A/H2B)$_2$ and purified by size exclusion chromatography as previously described (*Dyer et al., 2004*). Nucleosomes were generated by combining either the histone octamer or H2A/H2B dimer and H3/H4 tetramer (2:1 ratio) with DNA containing the Widom 601 sequence (*Lowary and Widom, 1998*). To favor hexasome formation, dimer and tetramer were combined in 1.2 : 1 ratio. Reconstitution by salt dialysis was performed as described (*Luger et al., 1999*). Nucleosomes and hexasomes were purified to ≥95% homogeneity by separating different nucleosomal species and free DNA over a 7% native acrylamide column (60:1 acrylamide:bisacrylamide) using a BioRad Prep Cell (Model 491) or MiniPrep Cell apparatus.

Addition of H2A/H2B dimer to hexasome was carried out separately for each reaction. H2A/H2B dimer (stored in 10 mM Tris-HCl (pH 7.8), 2 M NaCl, 1 mM EDTA, 5 mM 2-mercaptoethanol) was diluted roughly 10–30 fold to 6 μM in reaction buffer. Dimer dilutions were performed just prior to experiments. Hexasome was added to the reaction buffer first, followed by dimer in the indicated molar ratio. Dimer addition was allowed to proceed at room temperature for 2–3 min before additional reaction components were introduced; time courses of dimer addition indicated that incorporation of the dimer into hexasomes was complete within 30 s (data not shown). A similar pre-incubation step was carried out for nucleosome-containing reactions.

## Native gel sliding

Nucleosome sliding reactions were carried out as previously described with some minor adjustments (*Eberharter et al., 2004*; *Patel et al., 2011*). Briefly, 150 nM of fluorescently labeled nucleosome (or hexasome) and 50 nM Chd1 were diluted and combined in slide buffer (20 mM HEPES (pH 7.8), 100 mM KCl, 5 mM MgCl$_2$, 0.1 mg/mL BSA, 1 mM DTT, 5% sucrose (w/v)) at room temperature. Reactions were started with the addition of 2.5 mM ATP and at each time point, 1 μL of the reaction was added into into 24 μL of fresh quench buffer (20 mM HEPES (pH 7.8), 100 mM KCl, 0.1 mg/mL BSA, 1 mM DTT, 5% sucrose (w/v), 5 mM EDTA, 125 ng/μL salmon sperm DNA (Invitrogen)) and placed on ice. To visualize reaction products, 2.5 μL of the quenched time point samples were separated using 7% native polyacrylamide gels (60:1 acrylamide to bis-acrylamide) that were electrophoresed (125 V) for 2 hr at 4°C. Reaction products were observed by their fluorescent labels using a Typhoon 9410 variable mode imager (GE Healthcare).

## Histone mapping and Chd1 cross-linking

Histone mapping and Chd1 cross-linking experiments were conducted as previously described (*Kassabov and Bartholomew, 2004*; *Nodelman et al., 2017*). For each, single cysteine residues on either the nucleosome (H2B-S53C) or Chd1 (N459C or V721C) were labeled with 200–400 μM 4-azidophenacyl bromide (APB) at room temperature and in the dark for 2–3 hr and then quenched with DTT. For histone mapping, 150 nM APB-labeled nucleosomes were incubated with 50 nM Chd1 in slide buffer (20 mM Tris-HCl (pH 7.8), 50 mM KCl, 5 mM MgCl$_2$, 5% sucrose (w/v), 0.1 mg/mL BSA, 1 mM DTT). Sliding reactions were initiated with the addition of 2 mM ATP. At each time-point, 50 μL of the reaction was added to 100 μL of quench buffer (20 mM Tris-HCl (pH 7.8), 50 mM KCl, 5% sucrose (w/v), 0.1 mg/mL BSA, 5 mM DTT, 5 mM EDTA, 150 ng/μL salmon sperm DNA) and placed on ice. For Chd1 cross-linking, 300 nM APB-labeled Chd1 was incubated with 150 nM Cy3-40-601-40-FAM template DNA, nucleosomes, or hexasomes and 2 mM ADP BeF$_3$ (generated in each

reaction by adding 2 mM ADP, 15 mM NaF, 3 mM $BeCl_2$, and 6 mM $MgCl_2$) in slide buffer without additional $MgCl_2$. Incubations were carried out for 30 min in the dark at room temperature.

For both histone mapping and Chd1 crosslinking experiments, APB was crosslinked to the DNA by irradiating at 302 nm for 15 s using a UV Transilluminator (VWR). Samples were denatured with 0.1% SDS and heating to 70℃, and then subjected to phenol chloroform extraction and EtOH precipitation to remove uncrosslinked material. The crosslinked DNA was resuspended and cleaved with NaOH. The fragmented DNA was EtOH precipitated again, resuspended in formamide loading buffer, and separated on an 8 M urea, 8% polyacrylamide (19:1 acrylamide:bis-acrylamide) sequencing gel. The samples were run for 1.25 hr (1.75 hr for Chd1 crosslinking) at 65 W alongside a sequencing ladder of the nucleosomal DNA to allow precise identification of cross-link locations. Gels were imaged on a Typhoon 9410 variable mode imager (GE Healthcare) and analyzed using ImageJ (http://imagej.nih.gov/ij/).

## Exonuclease III digestion

ExoIII digestion was carried out on nucleosomes and hexasomes with fluorescently labeled DNA. Samples containing nucleosome (100 nM), hexasome alone (100 nM), or hexasome (100 mM) preincubated for 2–3 min with two fold molar excess H2A/H2B dimer were incubated at room temperature for 10 min in reaction buffer consisting of 20 mM HEPES (pH 7.6), 50 mM KCl, 10 mM $MgCl_2$, 5% sucrose (w/v), 0.1 mg/mL BSA, and 1 mM DTT. For each sample condition, four 10 μL digestion reactions were made containing 0, 10, 40, and 160 units of ExoIII (New England Biolabs). After digesting for 5 min at room temperature, reactions were quenched by the addition of 40 μL of quench buffer (20 mM HEPES (pH 7.6), 50 mM KCl, 20 mM EDTA, 1.2% SDS) and placed on ice. DNA was isolated from the digestion reactions by adding an equal volume of phenol:chloroform:isoamyl alcohol (25:24:1), vortexing, centrifuging for 2 min, and removing the top (aqueous) layer to a new tube. To completely remove phenol, this step was repeated using chloroform:isoamyl alcohol (24:1). DNA was precipitated by adding 1.5 μL of 10 mg/mL glycogen, 5 μL of 3 M sodium acetate and 250 μL 100% EtOH and then chilled at −80℃ for >20 min followed by centrifugation (21,130 rcf) for 30 min at 4℃. After a 70% EtOH wash and air drying of the pellet, samples were resuspended in 8 μL of formamide loading buffer and separated on urea sequencing gels as described for histone mapping.

## Single molecule FRET

Biotinylated and dye-labeled nucleosomes and hexasomes (alone or pre-incubated with an approximately twofold molar excess of unlabeled H2A/H2B dimer) were surface-immobilized on poly(ethylene glycol)-coated quartz microscope slides via a biotin-streptavidin linkage, as previously described (*Blosser et al., 2009*; *Deindl et al., 2013*). Immobilized samples were excited with a 532 nm Nd: YAG laser (CrystaLaser), and fluorescence emissions from Cy3 and Cy5 were detected using a prism-type TIRF microscope, filtered with a 550 nm long-pass filter (Chroma Technology), spectrally separated by a 635 nm dichroic mirror (Chroma Technology), and imaged onto the two halves of an Andor iXon Ultra 897 (512 × 512) CCD camera. The imaging buffer contained 12 mM HEPES, 40 mM Tris-HCl (pH 7.5), 60 mM KCl, 3 mM $MgCl_2$, 0.32 mM EDTA, 10% glycerol, an oxygen scavenging system (800 μg $ml^{-1}$ glucose oxidase, 40 μg $ml^{-1}$ catalase, 10% glucose) to reduce photobleaching, 2 mM Trolox (Sigma-Aldrich) to suppress photoblinking of the dyes (*Rasnik et al., 2006*), and 0.1 mg/ml BSA (Promega). Remodeling was induced by infusing the sample chamber with the imaging buffer containing 300 nM Chd1 remodeling enzyme and ATP using a syringe pump (Harvard Apparatus).

## Fluorescence experiments

Static quenching of fluorescence (SQOF) experiments were carried out using Cy3-Cy3 or Cy3B-Dabcyl pairs. Reactions were monitored for 0-601-80 nucleosomes or hexasomes, with exit-side H2A T120C labeled with Cy3 or Cy3B and the zero-end of DNA labeled with Cy3 or Dabcyl (IDT). Sliding reactions were conducted with 10 nM nucleosome or 10 nM hexasome with 12 nM dimer (unless otherwise noted), 25, 400, or 600 μM Chd1, and 25 μM ATP (except for *Figure 6* in which 1 mM ATP was used) ,100 mM KCl, 20 mM HEPES (pH 7.5), 5 mM $MgCl_2$, 100 μM EDTA, 5% sucrose w/v, 1 mM DTT, 0.2% Nonidet P-40, and 0.1 mg/mL BSA at 25℃.

Sliding reactions were monitored by either fluorometer or stopped-flow. Fluorometer experiments were conducted on a Fluorolog-3 fluorometer (Horiba) using a 2 mL reaction volume with a stir bar in the cuvette. First, 10 nM nucleosome or hexasome followed by H2A/H2B dimer was added to the cuvette and allowed to equilibrate for 2–3 min. Next, Chd1 was added, and after another brief equilibration, the sliding reaction was initiated with 25 μM ATP. Cy3 (or Cy3B) was excited at 510 nm and fluorescence was monitored at 565 nm using a 4 nm slit width and 1 s integration time. Stopped flow experiments were conducted on an SX20 stopped-flow (Applied Photophysics Limited) with nucleosome (or hexasome and dimer) and Chd1 in one syringe and ATP in the other. Cy3 (or Cy3B) was excited at 510 nm and emissions were monitored above 570 nm with a long-pass filter. Fluorescence signal was integrated over 0.01 s for the first ten seconds of the reaction and then 0.1 s for the remainder of the trace. Each progress curve is the average of 3–6 technical replicates. Progress curves were fit using the double exponential function, $yobs = a_1 \left(1 - e^{-k_1 \cdot x}\right) + a_2 \left(1 - e^{-k_2 \cdot x}\right)$ in Mathematica (Wolfram), where $k_1$ and $k_2$ are observed rates, $a_1$ and $a_2$ are corresponding amplitudes, and c is a constant.

## Acknowledgements

We thank Sarah Woodson for advice with stopped flow experiments and generously sharing equipment, Cynthia Wolberger and Mike Morgan for H2A-E64R protein and expression plasmids for Ubiquitin(G76C) and H2B(K117C), Geeta Narlikar and Nathan Gamarra for H2A-E61A/E64A/D90A/E92A protein, Sua Myong for suggesting use of Cy3B, and Ilana Nodelman for providing additional Chd1 protein, nucleosomes and hexasomes used for smFRET experiments, as well as N459C and V721C Chd1 variants. This work was funded by the National Institutes of Health (R01-GM084192 to GDB; T32-GM007231 to RFL) and the KAW Foundation and Swedish Research Council (SD).

## Additional information

### Funding

| Funder | Grant reference number | Author |
| --- | --- | --- |
| National Institutes of Health | R01-GM084192 | Gregory D Bowman |
| Vetenskapsrådet | | Sebastian Deindl |
| Knut och Alice Wallenbergs Stiftelse | | Sebastian Deindl |
| National Institutes of Health | T32-GM007231 | Robert F Levendosky |

The funders had no role in study design, data collection and interpretation, or the decision to submit the work for publication.

### Author contributions

RFL, Conceptualization, Resources, Formal analysis, Investigation, Methodology, Writing—original draft, Writing—review and editing; AS, Formal analysis, Investigation, Writing—review and editing; SD, Resources, Formal analysis, Supervision, Funding acquisition, Investigation, Writing—review and editing; GDB, Conceptualization, Formal analysis, Supervision, Funding acquisition, Methodology, Writing—original draft, Writing—review and editing

### Author ORCIDs

Gregory D Bowman, http://orcid.org/0000-0001-8025-4315

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
