## [Decision Letter]

Thank you for submitting your article "The Chd1 chromatin remodeler shifts hexasomes unidirectionally" for consideration by *eLife*. Your article has been favorably evaluated by Jessica Tyler as the Senior Editor and four reviewers, one of whom, Jerry L Workman (Reviewer #1), is a member of our Board of Reviewing Editors. The following individual involved in review of your submission has agreed to reveal their identity: Song Tan (Reviewer #2).

The reviewers have discussed the reviews with one another and the Reviewing Editor has drafted this decision to help you prepare a revised submission.

Summary:

This manuscript describes a novel method for producing hexasomes (nucleosomes containing one instead of two histone dimers) in a unique orientation. One would expect that the two histone dimers in the symmetric nucleosome would be equivalent, but the authors provide compelling evidence that the histone H2A/H2B dimers in nucleosomes reconstituted on the Widom 601 strong nucleosome positioning DNA sequence are not equivalent. In fact, the authors show that when histone tetramers, histone dimers and the Widom 601 positioning sequence are reconstituted with limiting amounts of histone dimer, the histone dimer is preferentially reconstituted on one side of the nucleosome (the TA-rich side). The authors further show that such oriented hexasomes can be converted into nucleosomes by incubation with additional histone dimer, thus providing a method for producing nucleosomes with labeled histone dimer in a specific orientation with respect to the nucleosomal DNA. Overall the manuscript includes some important observations, but is a little disjointed. The manuscript describes a new tool for chromatin biologists to perform targeted biochemical and spectroscopic experiments to shed mechanistic insights into the dynamics of the nucleosomes or how chromatin enzymes function on nucleosomes. It is difficult to assess how widely this will be used since there are a relatively small number of groups performing single molecule assays on nucleosomes. However, the ability to generate asymmetric nucleosomes may be useful in chromatin structural biology in ways that have not yet been thought of.

Essential revisions:

1) My major concern is with the interpretation that the experiments in Figure 2 show that Chd1 requires the entry side histone H2A/H2B dimer for sliding. How does one distinguish between a requirement for H2A/H2B on the entry side versus a requirement for particular DNA sequence on the entry side when the presence of H2A/H2B on the entry side is dependent on the entry side DNA sequence? I am not sure that one can easily deconvolute the role of H2A/H2B and the role of DNA sequence. The strongest piece of evidence in favor of the authors' interpretation is the stimulation of Chd1 by ubiquitylated H2B on the entry-side dimer (Figure 5). I encourage the authors to consider if they can devise experiments to analyze Chd1's requirement for H2A/H2B versus DNA sequence on the entry-side. If this is not possible in a timely manner, the authors should clearly discuss this issue in the manuscript.

2) I think it would be appropriate for the authors to comment or speculate on the molecular basis for the preferred deposition of the H2A/H2B dimer on the TA-rich side. This is particularly relevant given the question raised above of the potential difficulty distinguishing between the role of the entry-side H2A/H2B and the entry-side DNA sequence. One obvious possibility to discuss is the DNA sequence occupied by the H2A/H2B dimer in the oriented hexasome.

3) For Figure 1 presumably the hexasome species has been gel purified from a native gel. Otherwise the loss of dimers is higher than expected as the reconstitution contains a mix of nucleosomes and hexamers.

4) For Figure 1 chromatin assembly is most likely performed under conditions that favour hexasome formation. This should be stated clearly in the text and figure legend as for 1B a 1:1 stoichiometry of H3/H4:H2A/H2B is likely to have been used.

5) A convention is adopted where the side of the 601 sequence containing 4 TA steps is oriented to the right. Is this the orientation most researchers have adopted previously? If not, is there a good reason to use this orientation?

6) Figure 1, Figure 2, Figure 4, Figure 5: Markers should be indicated on the gels shown in the figures to allow easier comparison of the data between the figures. The text in the figures is too small and difficult to read.

7) Figure 1: Since the gel shown here is representative of three similar experiments using different hexasome/nucleosome preparations, the standard deviations should be shown on the bar graph.

8) Figure 1: The histone mapping experiment should be better described in the figure legend.

9) Figure 2: Using the scans, it would be important to quantitatively compare the relative efficiencies of nucleosome/hexasome translocation.

10) Using the fluorescence quenching approaches (Figure 4–Figure 5) it would be important to determine the step in the remodeling reaction (KM or kcat) that is affected by the removal of the H2A/H2B dimer on the entry side.

11) Figure 3, Figure 4: The experimental conditions used in the experiment should be described in the figure legend to allow easier comparison with data shown in the other figures.

12) Figure 4 is not essential and could be transferred to a figure supplement.

---

## [Author Response]

*Essential revisions:*

*1) My major concern is with the interpretation that the experiments in Figure 2 show that Chd1 requires the entry side histone H2A/H2B dimer for sliding. How does one distinguish between a requirement for H2A/H2B on the entry side versus a requirement for particular DNA sequence on the entry side when the presence of H2A/H2B on the entry side is dependent on the entry side DNA sequence? I am not sure that one can easily deconvolute the role of H2A/H2B and the role of DNA sequence. The strongest piece of evidence in favor of the authors' interpretation is the stimulation of Chd1 by ubiquitylated H2B on the entry-side dimer (Figure 5). I encourage the authors to consider if they can devise experiments to analyze Chd1's requirement for H2A/H2B versus DNA sequence on the entry-side. If this is not possible in a timely manner, the authors should clearly discuss this issue in the manuscript.*

The reviewer brings up an important point that we had not considered in the previous version of the manuscript. We realized that a direct answer to this question is shown in Figure 4, where 80-601-0 hexasomes show a different distribution than 80-601-0 nucleosomes. Specifically, hexasomes are shifted farther to the opposite end of the DNA, which means that the histone core has moved ~75 bp from its starting position and therefore changed the sequence of DNA over the location where the missing H2A/H2B dimer would sit. The altered distribution of the hexasome toward the side with the entry side H2A/H2B dimer indicates that the histone core is not readily shifted toward the side lacking the dimer, even when the DNA sequence over the missing dimer is completely different. We therefore conclude that it is the lack of the H2A/H2B dimer and not the sequence that is responsible for the poor sliding of hexasomes with the dimer missing on the entry side. This conclusion is presented in the Results: “This biased movement of 80-601-0 toward the entry H2A/H2B dimer, even after the hexasome had shifted away from its starting position on the 601 sequence, was consistent with much poorer sliding toward the side lacking the H2A/H2B dimer and indicated that it was the absence of H2A/H2B rather than the DNA sequence that blocked efficient sliding of 0-601-80 hexasomes.”

*2) I think it would be appropriate for the authors to comment or speculate on the molecular basis for the preferred deposition of the H2A/H2B dimer on the TA-rich side. This is particularly relevant given the question raised above of the potential difficulty distinguishing between the role of the entry-side H2A/H2B and the entry-side DNA sequence. One obvious possibility to discuss is the DNA sequence occupied by the H2A/H2B dimer in the oriented hexasome.*

As we now describe in the Discussion, we favor the idea that the periodic TA steps are responsible for the marked asymmetry in H2A/H2B deposition. Previous work by the Davey group clearly showed that the TA-rich side of the 601 is much more salt tolerant than the TA-poor side. In their Chua et al. (2012, NAR) paper, the authors generated symmetric 601 sequences, with two copies of the TA-rich side (called 601L) showing significantly higher salt stability compared with 601R, made with two copies of the TA-poor side. While it may very well be the ~30 bp sequence that directly contacts H2A/H2B that is critical, we believe that the force-unwrapping paper by the Ha lab (Ngo et al., 2015, Cell) demonstrates that interactions between H3/H4 and DNA around the dyad can influence stability of adjacent H2A/H2B-DNA contacts farther (~30 bp) from the dyad. However, since one of the four TA step positions (at SHL 3.5) also contacts the H2A/H2B dimer, future experiments correlating the orientation of the hexasome with DNA sequence will be required to tease apart direct versus indirect sequence effects.

*3) For Figure 1 presumably the hexasome species has been gel purified from a native gel. Otherwise the loss of dimers is higher than expected as the reconstitution contains a mix of nucleosomes and hexamers.*

We apologize for this confusion. We somehow failed to articulate that all of the nucleosomes and hexasomes are purified away from each other, which has allowed us to study the properties of each separately. To help make this point clearer, we now include a purification gel and show the separated nucleosome and hexasome pools in Figure 1.

*4) For Figure 1 chromatin assembly is most likely performed under conditions that favour hexasome formation. This should be stated clearly in the text and figure legend as for 1B a 1:1 stoichiometry of H3/H4:H2A/H2B is likely to have been used.*

The reviewer is correct – to enrich hexasomes, we preferentially use a 1:1 mixture of H2A/H2B dimer with (H3/H4)_2_ tetramer. As mentioned above, we now describe this more specifically in the manuscript (Figure 1, Results, and Methods).

*5) A convention is adopted where the side of the 601 sequence containing 4 TA steps is oriented to the right. Is this the orientation most researchers have adopted previously? If not, is there a good reason to use this orientation?*

We agree with the suggestion of the reviewer, and for clarity now present the 601 sequence in the orientation that matches several other publications that explicitly describe the TA steps.

*6) Figure 1, Figure 2, Figure 4, Figure 5: Markers should be indicated on the gels shown in the figures to allow easier comparison of the data between the figures. The text in the figures is too small and difficult to read.*

The revised figures now include sequencing ladders for the histone mapping and cross-linking experiments. For native gels, ladders are not typically used; as we show in Figure 1, the different pools are analyzed separately by SDS-PAGE to confirm their histone content. From many purifications and biochemical experiments, migration of hexasomes, nucleosomes, and free DNA in native gels is predictable, and we show in Figure 1 how the purified pools of hexasomes and nucleosomes matches with the pre-purified reconstitution (load), run on the same gels. To illustrate the differences in hexasome migration patterns, we now also show a native gel where the two hexasome samples (0-601-80 and 80-601-0) are loaded side-by-side (Figure 4—figure supplement 1).

We have reformatted the text and sizing of figures to improve readability. This reformatting required that we split Figure 1 and Figure 2 into several figures in the revised manuscript.

*7) Figure 1: Since the gel shown here is representative of three similar experiments using different hexasome/nucleosome preparations, the standard deviations should be shown on the bar graph.*

We now show standard deviations for this bar graph, which is presented as Figure 1.

*8) Figure 1: The histone mapping experiment should be better described in the figure legend.*

Figure 1 in the original submission now corresponds to Figure 3 in the revised manuscript. We have added a brief description of histone mapping in the figure legend and main text, as well as in the Methods section. We have also made an effort to include more methodological explanations in other figure legends.

*9) Figure 2: Using the scans, it would be important to quantitatively compare the relative efficiencies of nucleosome/hexasome translocation.*

In this figure (which is Figure 4 in the revised manuscript), we include a “% remaining” under the intensity plot for each experiment, which gives the fraction of material that appears to have not shifted away from the starting location. To obtain a more quantitative comparison of sliding rates, we now include a real-time fluorescence experiment that provides the kinetic parameters for nucleosome versus hexasome sliding (Figure 6 and Figure 6—figure supplement 1).

*10) Using the fluorescence quenching approaches (Figure 4–Figure 5) it would be important to determine the step in the remodeling reaction (KM or kcat) that is affected by the removal of the H2A/H2B dimer on the entry side.*

We address this issue in two ways. First, we provide new cross-linking data that shows the Chd1 ATPase motor engages equally well with SHL2 DNA for both nucleosomes and hexasomes (Figure 6). Second, we perform nucleosome and hexasome sliding reactions with saturating amounts of Chd1, which is unable to recover the sliding defect (Figure 6 and Figure 6—figure supplement 1). These results point to the sliding defect being a reduction in kcat.

*11) Figure 3, Figure 4: The experimental conditions used in the experiment should be described in the figure legend to allow easier comparison with data shown in the other figures.*

These are now Figure 5 and Figure 7—figure supplement 1; more experimental detail has been included in these legends.

*12) Figure 4 is not essential and could be transferred to a figure supplement.*

We agree with the reviewer and now present these data as Figure 7—figure supplement 1.